# Detecting Land Degradation in Eastern China Grasslands with Time Series Segmentation and Residual Trend analysis (TSS-RESTREND) and GIMMS NDVI$_{3g}$ Data

**Caixia Liu** [1,2,]**\***, **John Melack** [2], **Ye Tian** [1], **Huabing Huang** [1], **Jinxiong Jiang** [3], **Xiao Fu** [4] and **Zhouai Zhang** [5,6]

1   State Key Laboratory of Remote Sensing Science, Jointly Sponsored by the Institute of Remote Sensing and Digital Earth of Chinese Academy of Sciences and Beijing Normal University, Beijing 100101, China; tianye@radi.ac.cn (Y.T.); huanghb@radi.ac.cn (H.H.)
2   Bren School of Environmental Science and Management, University of California, Santa Barbara, CA 93106, USA; melack@lifesci.ucsb.edu
3   State Key Laboratory of Space-Ground Integrated Information Technology, Beijing 100029, China; jiangjx@spacestar.com.cn
4   State Key Laboratory of Urban and Regional Ecology, Research Center for Eco-Environmental Sciences, Chinese Academy of Sciences, Beijing 100085, China; xiaofu@rcees.ac.cn
5   Shenhua Baorixile Energy Co., Ltd., Hulunbuir 021000, China; 11550128@chnenergy.com.cn
6   State Key Laboratory of Water Resource Protection and Utilization in Coal Mining, Beijing 100011, China
\*   Correspondence: caixialiu@ucsb.edu

**Abstract:** Grassland ecosystems in China have experienced degradation caused by natural processes and human activities. Time series segmentation and residual trend analysis (TSS-RESTREND) was applied to grasslands in eastern China. TSS-RESTREND is an extended version of the residual trend (RESTREND) methodology. It considers breakpoint detection to identify pixels with abrupt ecosystem changes which violate the assumptions of RESTREND. With TSS-RESTREND, in Xilingol (111°59′–120°00′E and 42°32′–46°41′E) and Hulunbuir (115°30′–122°E and 47°10′–51°23′N) grassland, 6% and 3% of the area experienced a decrease in greenness between 1984 and 2009, 80% and 73% had no significant change, 5% and 3% increased in greenness, and 9% and 21% were undetermined, respectively. RESTREND may underestimate the greening trend in Xilingol, but both TSS-RESTREND and RESTREND revealed no significant differences in Hulunbuir. The proposed TSS-RESTREND methodology captured both the time and magnitude of vegetation changes.

**Keywords:** grassland; NDVI; RESTREND; BFAST; land degradation

## 1. Introduction

Grasslands are the largest ecosystem in China, covering 393,000 km$^2$, 43% of the country territory [1]. In the past forty decades, the grasslands in China have experienced serious land degradation caused by natural process and human activities, including climate change, land use alternations, and socioeconomic transformation [1,2]. Discerning the impacts of these drivers is important for understanding and managing landscapes, particularly for grasslands in arid and semi-arid areas. In these areas, annual precipitation is low and inter-annual variability in rainfall high [3,4]. When conducting vegetation greenness change and land degradation analysis in these areas caused by human activities, the first step is to exclude the impact from climatic variations. The climate

variability makes it difficult to separate natural changes in vegetation from those caused by direct human activities [5–7]. Given that land degradation may occur progressively over many decades, it is necessary that measurements be consistent, and combined with large areas influenced [5,8], remote sensing is an appropriate approach.

Several studies of the relations between the normalized difference vegetation index (NDVI) and climatic factors have been used to separate changes in vegetation caused by climate from those caused by both anthropogenic and natural factors [8–10]. The residual trend approach (RESTREND) [11] is a method for removing the climate influence from an NDVI trend, and it has been used for detection of dryland degradation based on climatic data and vegetation indices [5,11]. It was initially developed to control for variations in climate influencing vegetation indices by calculating a linear regression between annual maximum NDVI, a proxy for ecosystem productivity, and precipitation [6,11]. The difference between the observed NDVI and estimated NDVI from the linear regression, referred to as the NDVI residuals, was then calculated, and linear trend analysis implemented using the NDVI residuals. However, RESTREND has a limitation because it provides valid results only when there is a strong linear relationship between the variations in the precipitation and vegetation index. When degradation occurred in the middle of the time series, a strong linear relationship may be lacking and lead to unreliable results [7]. Thus, it is of importance to detect if degradation causes the linear relationship between vegetation and precipitation to breakdown (also termed as a breakpoint). Breakpoint detection on the time series of remotely sensed products has been applied to land cover mapping [12–14] and forest management [15,16]. It has been argued that dryland degradation could be improved when detecting breakpoint when the relationship between vegetation and precipitation breaks down [5,7,17]. Introduced by Burrell, Evans, and Liu [8], time series segmentation and residual trend analysis (TSS-RESTREND) is such a methodology, and combines RESTREND time series analysis with breakpoint detection using breaks for additive seasonal and trend (BFAST) [18,19]. It was successfully used to detect land degradation in Australia [8,10] and was able to improve the detection of degraded areas and direction of change compared to RESTREND alone. The recent study proposed by Abel et al. [17] also used BFAST for breakpoint detection on sequential linear regression slopes (SeRGS) for time series rainfall and vegetation relationship.

In the eastern grasslands of China, most studies focus on the Mongolian plateau [20–22] or Xilingol grasslands [9,23,24]. Our study areas are the eastern grasslands of Xiliingol and Hulunbuir, located in arid and semi-arid regions experiencing decreased rainfall [25] and frequent human activities [23,24,26]. Hulunbuir and Xilingol have many coal-fired power plants [27]. Coal production in Inner Mongolia accounted for 25% of China's production. The mining of coal and construction of power plants may degrade vegetation by reducing surface and subsurface water resources as well as pollution from the power plants [28,29]. To the best of our knowledge, no studies have examined and compared land degradation in Xilingol and Hulunbuir grasslands using a RESTREND or a similar methodology to control for variations in precipitation.

The aim of this study is to develop an understanding of land degradation in the eastern grasslands of China for the period from 1984 to 2009. Xilingol and Hulunbuir grasslands were chosen as study areas due to their representative roles as typical steppe and meadow steppe in the arid and semi-arid area of eastern China. The specific objectives in our study area are: (1) to test the new TSS-RESTREND method, (2) to explore breakpoints in NDVI time series, and (3) to compare the performance of TSS-RESTREND results with other statistical methods.

## 2. Materials and Methods

### 2.1. Study Area

Inner Mongolian grasslands are regarded as an important ecological barrier to the north of China and East Asia because the vegetation can reduce the effects of wind and dust when the winter northwest monsoon passes through. These grasslands extend more than 3000 km from east to west, with the

grasslands gradually changing from meadow steppe to steppe, desert steppe, and desert [30,31]. The total area of grasslands is 749,000 km$^2$, accounting for 63% of Inner Mongolia. Most (84%) of the grasslands are used for grazing [32]. Hulunbuir and Xilingol in Figure 1, were chosen for study because they are the main land cover types in our study areas.

The Xilingol grasslands (111°59′–120°00′E and 42°32′–46°41′E) are situated in the middle part of the Inner Mongolia Autonomous Region, northeastern China (Figure 1B). The elevation of Xilingol is between around 750 and 1900 m. The terrain has low hills in the east and south and is flat in the west and north, with a sporadic distribution of low hills and lava platforms. It is characterized by a mid-temperate semi-arid continental climate type. The average annual precipitation ranges from 135 mm to 380 mm. Xilingol grasslands are considered to be the most complete wild grassland in the grassland subzone of East Asia in the Eurasian steppe region, with an area of approximately 200,000 km$^2$, of which the grassland area accounts for 98%. The main vegetation types include various formations of desert steppes, typical steppes, and meadow steppes (Figure 2).

The area of Hulunbuir grasslands (115°30′–122°E and 47°10′–51°23′N) is approximately 83,000 km$^2$. The topography of Hulunbuir becomes gradually flatter with a decrease in elevation from the center to the east and west. The elevation of Hulunbuir ranges from about 170 to 1700 m. It has a typical temperate continental monsoon climate, with the annual precipitation approximately 250 mm to 400 mm; average annual temperature is approximately −3 °C to 0 °C. The study area is dry and windy in spring and warm in summer. Temperatures decrease quickly in autumn, and winter is long and cold. The main grassland vegetation in the region is meadow steppe and steppe from east to west (Figure 2).

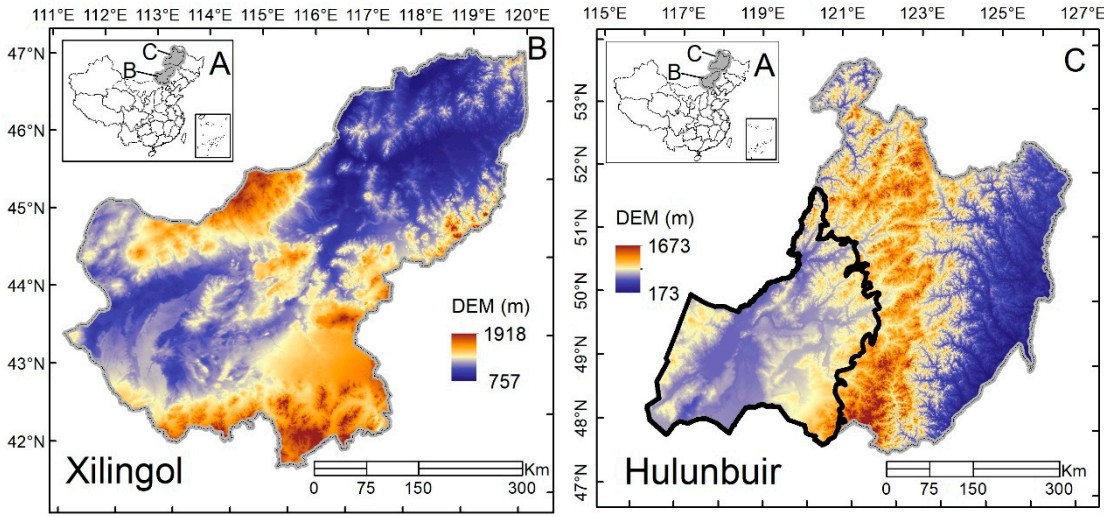

**Figure 1.** Elevation of the study area. The location of the study area in China is shown (**A**). The topographic elevations of Xilingol and Hulunbuir are shown (**B**,**C**), respectively. Grasslands are demarcated with a dark line (**C**).

## 2.2. Data Source and Pre-Processing

### 2.2.1. GIMMS NDVI$_{3g}$ Data

The global inventory modeling and mapping studies (GIMMS) 16-day composite NDVI3g dataset was used for vegetation change. GIMMS NDVI3g data are currently the longest time series of NDVI for monitoring the characterization and variability of vegetation. Yin et al. [33] argued the importance of temporally consistent NDVI for vegetation monitoring and some studies have shown that NDVI3g is more accurate than the GIMMS NDVI for monitoring vegetation activity and phenological change [34]. GIMMS NDVI3g data were derived from the AVHRR instrument onboard the NOAA satellite series (7, 9, 11, 14, 16–19) from July 1981 to December 2013. The dataset has been corrected for calibration, solar geometry, aerosols, clouds, and other effects not related to vegetation change [35]. Monthly

GIMMS NDVI was computed from the GMMIS-NDVI3g with 16-day temporal resolution through the maximum-value composite procedure (MVC) [36]. We used cloud-based Google Earth Engine to process this dataset (imageCollection ID is NASA/GIMMS/3GV0) using quality record (QA flags) in each time series image to filter for high quality monthly maximum NDVI data.

### 2.2.2. Meteorological Dataset

The National Meteorological Information Center of China provided the monthly precipitation and temperatures from 1980 (http://cdc.cma.gov.cn/home.do). The monthly precipitation maps were produced with the thin-plate spline spatial interpolation provided by [37,38] and transformed to raster images with a resolution of 1 km; this rainfall dataset has been used for the analysis of changes in lake areas [39]. We did a projection transformation and spatially resampled to match the GIMMS NDVI images.

### 2.2.3. Vegetation Types

Vegetation spatial distribution is shown in Figure 2. The dataset was originally produced by the Institute of Botany, the Chinese Academy of Sciences [31], and we obtained it from Data Center for Resources and Environmental Sciences, Chinese Academy of Sciences (RESDC) (http://www.resdc.cn). The main vegetation in our study area using the classification of Hou [28] is temperate steppe, except for large areas of forest in the east of Hulunbuir. In Xilingol, desert steppe, typical steppe, and meadow steppe located from west to east. There are two steppe types in Hulunbuir: typical steppe and meadow steppe. We did not analyze areas outside the polygon which are mainly forests and croplands.

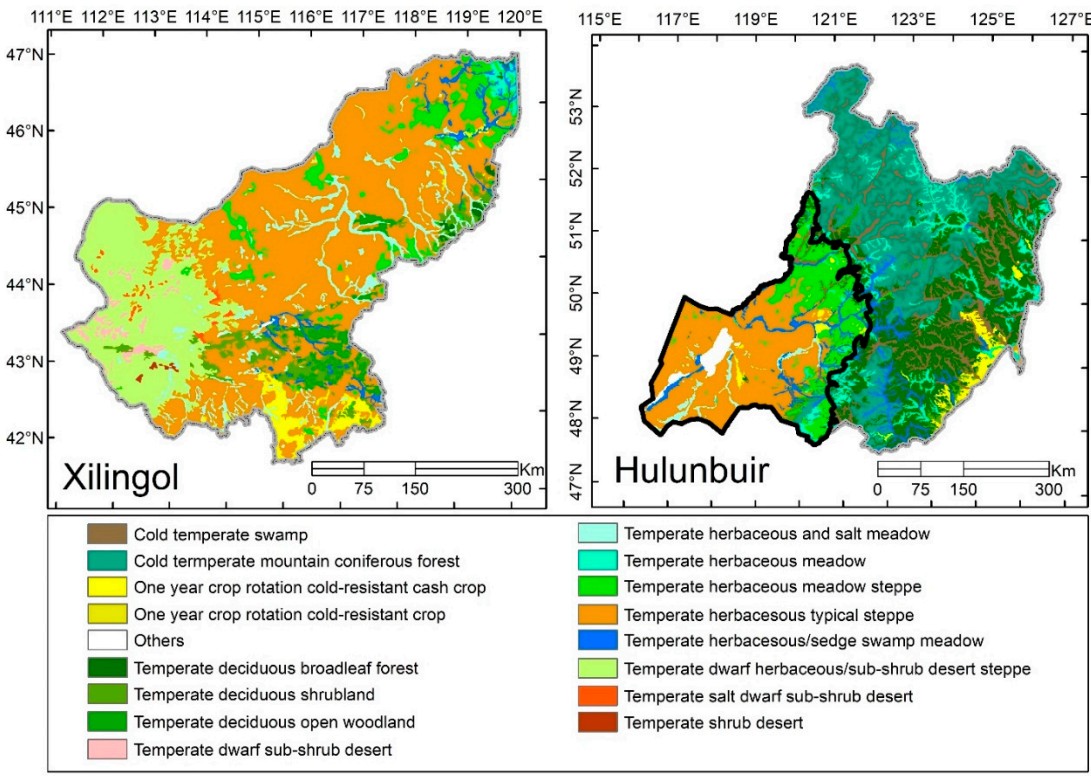

**Figure 2.** Vegetation types in the study area (reproduced according to Hou [28]).

### 2.2.4. Validating Breakpoint Timing in Google Earth Engine

For each pixel with a breakpoint detected by the BFAST method, time series Landsat images (1984–2009) were visually interpreted, and disturbance years were collected using a tool built under GEE. With this tool, the time series of NDVI can be plotted and a colored Landsat image can be

displayed if a designated date is chosen. For the displayed NDVI, the mean value was calculated on the basis of all Landsat pixels that are covered by the changed GIMMS pixel. The codes for the tool can be found at (https://code.earthengine.google.com/22e70048312b89d26ba6c2effd946caa).

*2.3. Methods*

2.3.1. RESTREND

RESTREND is a pixel-based approach developed by Evans and Geerken [11] and further modified by Wessels et al. [7] for distinguishing between natural variability and degradation processes in water-limited ecosystems [5,40]. It calculates the relationship between the net primary productivity (NPP) of vegetation and precipitation; hereafter we refer to this relationship as VPR (vegetation-precipitation relationship). An ordinary least squares regression (OLSR) is calculated between peak growing season NDVI (NDVImax), a proxy for NPP [7,8], and the optimal accumulated precipitation. In our study area, the peak growing season NDVI is equal to the annual maximum NDVI which occurred in summer. Optimal accumulation of precipitation is calculated pixel by pixel to determine the combination of accumulation period (1–12 months) and offset period (1–3 months) that produced the highest correlation coefficient with NDVImax. Predicted NDVImax is acquired by the OLSR model, and the difference between predicted NDVImax and observed NDVImax at each time is referred to as VPR-Residual.

Residual trend (RESTREND) of the VPR is then calculated by OLSR from the regression of the VPR-Residuals with time [11].

$$y_i = \beta_0 + \beta_1 x_i \tag{1}$$

where $y_i$ is VPR-Residuals, $x_i$ is years, $\beta_0$ is intercept and $\beta_1$ is slope.

Trends present in the VPR-Residuals are independent of precipitation and an indicator of the initiation or reversal of land change processes [11]. To apply RESTREND for analysis, a pixel must meet three criteria [8]:

i.    The VPR is significant and positive (slope > 0). Recommended values for significance are R2 > 0.3 at $p < 0.05$ significance level.
ii.   A residual trend is gradual and consistent or at least monotonic for the entire time series [41].
iii.  The VPR must remain consistent with time, which is defined as a VPR that is comparable throughout the entire time series, i.e., no major structural changes occurring within the ecosystem [7].

Hence, a standard RESTREND may fail to identify a trend when the rate and direction of change vary within the time series [42]. Previous studies excluded those pixels when they cannot meet those criteria [7,43].

2.3.2. TSS-RESTREND

The time series segmentation and residual trend (TSS-RESTREND) method was first proposed by Burrell, Evans, and Liu [8] for analysis of vegetation trends in Australia. The main purpose of TSS-RESTREND is to locate breakpoints of the time series of NDVI with the BFAST method [18,19]. Unlike the previous studies with BFAST [15,44], BFAST in TSS-RESTREND is not directly implemented on time series NDVI but on complete VPR residuals, in which the seasonal component of BFAST is switched off.

In each time segmentation, the Chow test [45] to the VPR-residuals is done to test the breakpoints' impact on the primary productivity, represented by NDVImax. Chow test is based on the null hypothesis that there is no change in the regression coefficient across a potential breakpoint and is rejected when the F-statistic reaches a critical threshold ($\alpha = 0.05$).

There are four situations when using TSS-RESTREND:

i.     If a pixel has a significant VPR ($\alpha = 0.05$) and no significant breakpoints in the VPR-residuals ($\alpha = 0.05$), it meets all the criteria for a standard RESTREND.

ii.    If a significant breakpoint is detected in the VPR-Residuals, a Chow test is also applied to the VPR. For a pixel with a significant breakpoint in the VPR-Residuals ($\alpha = 0.05$) but not in the VPR ($\alpha = 0.05$), a segmented RESTREND is applied, as shown in Burrell et al. [8], in which a multivariate regression between the VPR-Residuals, time and a dummy variable that is 0 before the breakpoint and 1 after it:

$$y_i = \beta_0 + \beta_1 x_i + \beta_2 z_i + \beta_3 x_i z_i \tag{2}$$

where $x$ = years, $z$ = value of the dummy variable (0 or 1). $\beta_0$ is intercept, $\beta_1$ is slope, $\beta_2$ is the offset at the breakpoint and $\beta_3$ = the change in the slope at the breakpoint.

iii.   If a pixel has a significant breakpoint in VPR, it may indicate a significant structural change to the ecosystem during the study period [46]. Therefore, it is not valid to assume that the optimal duration of precipitation is equivalent to either side of the breakpoint. Thus, the time series NDVImax is separated and a new VPR is recalculated separately on either side of the breakpoint. In order to make it possible to compare with different accumulation and offset periods across the breakpoint, Burrell et al. [8] converted the optimal precipitation into a standard score:

$$z_i = \frac{x_i - \mu}{\sigma} \tag{3}$$

where $z$ = standard score, $x_i$ = observed values, $\mu$ = mean of that accumulation period for the entire time series, $\sigma$ = standard deviation. Then a multivariate regression is fitted to the time series standard scores:

$$y_i = \beta_0 + \beta_1 x_i + \beta_2 z_i + \beta_3 x_i z_i \qquad i = 1984, \dots, 2009 \tag{4}$$

where $x$ = the standardized precipitation in formula (3) for year $i$, $z$ = value of the dummy variable (0 or 1), $\beta_0$ is intercept, $\beta_1$ is slope, $\beta_2$ is the offset at the breakpoint and $\beta_3$ is the change in the slope at the breakpoint. The total change of a pixel with a segmented VPR is calculated by adding the residual change to the VPR break height ($\beta_2$). The residual change is calculated with segmented RESTREND.

iv.   Pixels, where no significant model can be fitted, are classified as indeterminate.

Burrell et al. [8] gave a detailed algorithm description in their paper with a flow-chart; there is a small typo in Burrell et al. [8], and we corrected this typo here that affected Equation (3).

### 2.3.3. Linear Trend Analysis (LTA)

Linear trend analysis (LTA) is one of the most widely used approaches to monitor vegetation change using time series NDVI (NDVIts) data because it is a simple, intuitive way to identify continuous inter-annual vegetation change trends [24,47]. However, applying this method over long time series may be misleading as contrasting trends can potentially balance [5]. A trend in the raw LTA of NDVIts was calculated with OLSR without considering precipitation influence.

$$y_i = \beta_0 + \beta_1 x_i \tag{5}$$

where $y_i$ is NDVIts and $x_i$ is year. Pixels with $\beta_0 > 0$ are considered as a pixel in which vegetation increased productivity and $\beta_0 < 0$ as decreased productivity. The significance of the model was also recorded and combined with slope ($\beta_0$) to categorize pixels into different clusters indicating vegetation change direction and magnitude as described by Li et al. [43].

### 2.3.4. Comparison RESTREND, TSS-RESTREND, and LTA Results

We compared vegetation change direction and magnitude of standard RESTREND, TSS-RESTREND, and traditional LTA as described by Li et al. [43]. Pixels were categorized into nine classes by the direction of the change and its significance as shown by thresholds in Table 1.

**Table 1.** Category threshold for vegetation change.

| Category ID | Change Direction | Significance |
|:---:|:---:|:---:|
| I1 | | $\alpha < 0.01$ |
| I2 | slope > 0 | $0.01 \le \alpha < 0.025$ |
| I3 | | $0.025 \le \alpha < 0.05$ |
| INC | | $0.05 \le \alpha < 0.1$ |
| D1 | | $\alpha < 0.01$ |
| D2 | slope < 0 | $0.01 \le \alpha < 0.025$ |
| D3 | | $0.025 \le \alpha < 0.05$ |
| DNC | | $0.05 \le \alpha < 0.1$ |
| NSC | | $\alpha > 0.1$ |

## 3. Results

### 3.1. Vegetation Change Detection by TSS-RESTREND, RESTREND, and LTA

For TSS-RESTREND, 80% and 73% of the pixels were detected as unchanged in Xilingol and Hulunbuir, respectively (Tables 2 and 3). For those pixels considered unchanged with TSS-RESTREND, 80% in Xilingol and 87% in Hulunbuir had decreased based on the LTA results. Hence, these led to only 0.5% and 1% pixels considered as unchanged with LTA in Xilingol and Hulunbuir, respectively (Tables 2 and 3). Because TSS-RESTREND and RESTREND methods consider annual rainfall variations with greenness in arid areas and some pixels may fail to satisfy the relationship between NDVI and rainfall values, there are still undetermined areas with the two methods when analyzing relationships between rainfall and NDVI. In 9% (Xilingol) and 21% (Hulubuir) of the areas, trends cannot be detected if we consider the climatic impact on greenness with TSS-RESTREND.

**Table 2.** NDVI change results between TSS-RESTREND and LTA in Xilingol.

| TSS_RESTREND | | | | | |
|:---:|:---:|:---:|:---:|:---:|:---:|
| **No Change** | | | **Decrease** | **Increase** | **Undetermined** |
| 80% | | | 6% | 5% | 9% |
| **LTA** | | | | | |
| **No change** | **Decrease** | **Increase** | | | |
| 0.5% | 80% | 19.5% | | | |

| LTA | | | | | |
|:---:|:---:|:---:|:---:|:---:|:---:|
| **No change** | **Decrease** | | | **Increase** | **Undetermined** |
| 0.5% | 75.5% | | | 24% | 0 |
| | **TSS_RESTREND** | | | | |
| | **No Change** | **Decrease** | **Increase** | **Undetermined** | |
| | 84.8% | 7% | 8% | 0.2% | |

**Table 3.** NDVI changes results between TSS-RESTREND and LTA in Hulunbuir.

| TSS_RESTREND | | | | | | |
|---|---|---|---|---|---|---|
| No Change | | | Decrease | Increase | Undetermined | |
| 73% | | | 3% | 3% | 21% | |
| LTA | | | | | | |
| No change | Decrease | Increase | | | | |
| 1% | 87% | 12% | | | | |
| LTA | | | | | | |
| No change | Decrease | | | Increase | Undetermined | |
| 1% | 81% | | | 18% | 0 | |
| TSS_RESTREND | | | | | | |
| No Change | Decrease | Increase | Undetermined | | | |
| 78.3% | 4% | 17% | 0.7% | | | |

Changes in vegetation based on TSS-RESTREND and LTA reveal that an increase in NDVI occurred in northwestern and southwestern Xilingol and a decrease occurred in the northeast (Figure 3). The changed areas derived from TSS-RESTREND are less than those from LTA, but the spatial distribution of change is similar. In Hulunbuir, degraded areas occurred mainly in the western and southeastern areas based on LTA, whereas TSS-RESTREND suggested degraded areas were distributed in the central and southeastern areas.

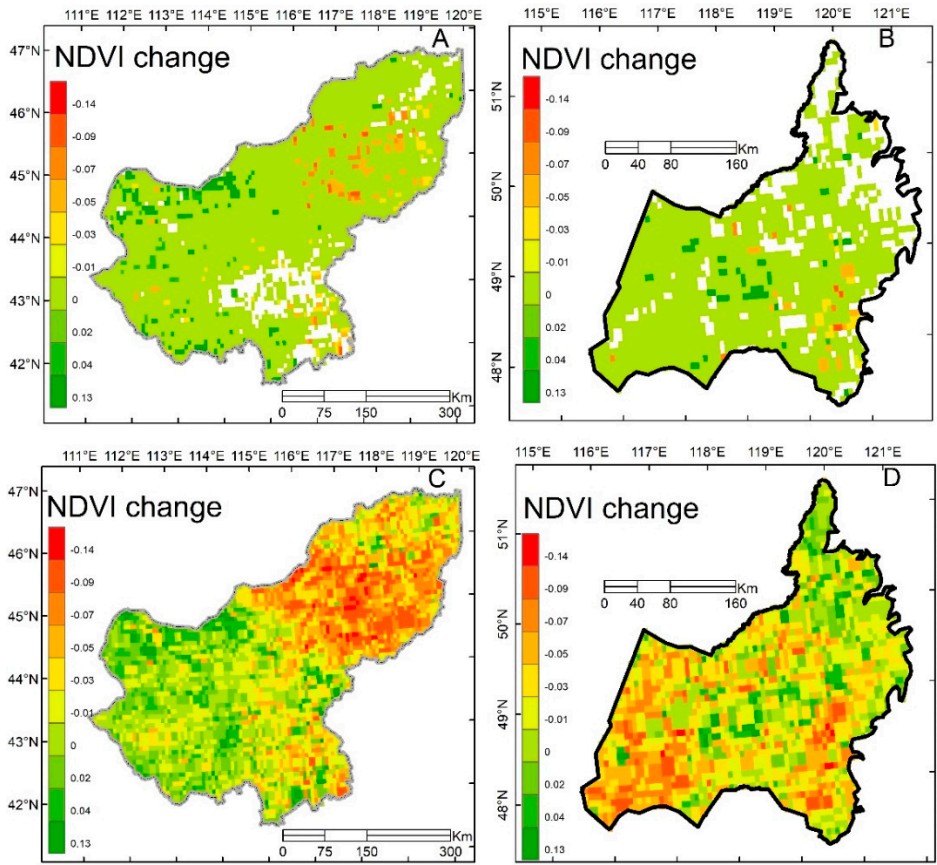

**Figure 3.** Vegetation change direction. (**A**,**B**) NDVI change trends from 1984–2009 in Xilingol and Hulunbuir with the TSS-RESTREND method. (**C**,**D**) The results with the LTA. The redder the color, the larger the decrease of NDVI. The greener the color, the larger the greenness change.

### 3.2. Trend and Breakpoint Analyses with TSS-RESTREND

From 1984 to 2009 NDVI in most areas of both study areas remained unchanged based on TSS-RESTREND (Figure 4). Following the rules in Table 1, severe degradation (D1, D2, and D3) occurred in northeastern and southeastern Xilingol (Figure 4A) and southeastern Hulunbuir (Figure 4B). Greening areas (I1, I2, and I3) are in the northwestern Xilingol and central Hulunbuir. TSS-RESTREND does not work well in croplands, meadow steppe, and deciduous shrublands or forests (Figure 5).

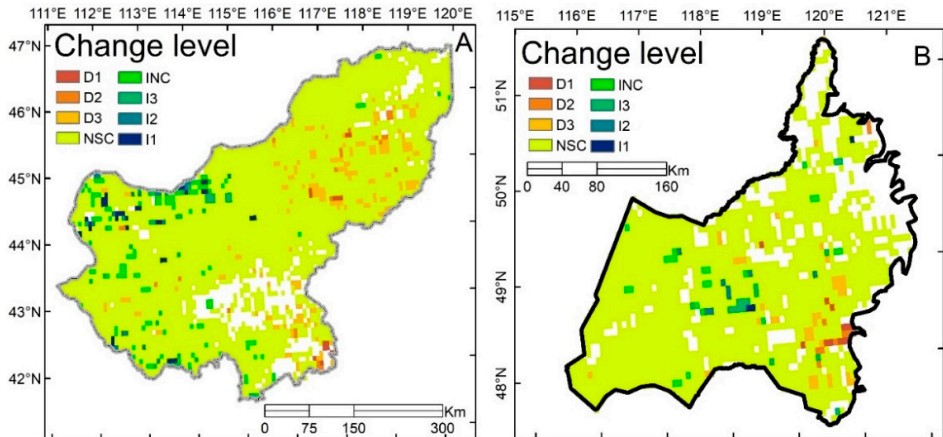

**Figure 4.** NDVI change direction (decrease/increase/no change) determined by TSS-RESTREND. (**A**: Xilingol; **B**: Hulunbuir). We divided the significance of the linear regression of residual and time into four levels (0.01, 0.025, 0.05, and 0.1) with the F-test and classified nine variations in residual trends. The decreasing trend are D1 ($p < 0.01$), D2 ($0.01 \leq p < 0.025$), D3 ($0.025 \leq p < 0.05$), and DNC ($0.05 \leq p < 0.1$), and the increasing trend are I1 ($p < 0.01$), I2 ($0.01 \leq p < 0.025$), I3 ($0.025 \leq p < 0.05$), and INC ($0.05 \leq p < 0.1$). D1, D2, and D3 indicate obvious decreases in vegetative productivity, whereas I1, I2, and I3 represent increases. DNC and INC represent observable decreasing and increasing trends, respectively, with statistical significance between 0.05 and 0.1. The NSC refers to an insignificant statistical relationship in the trend of residual variation ($p > 0.1$).

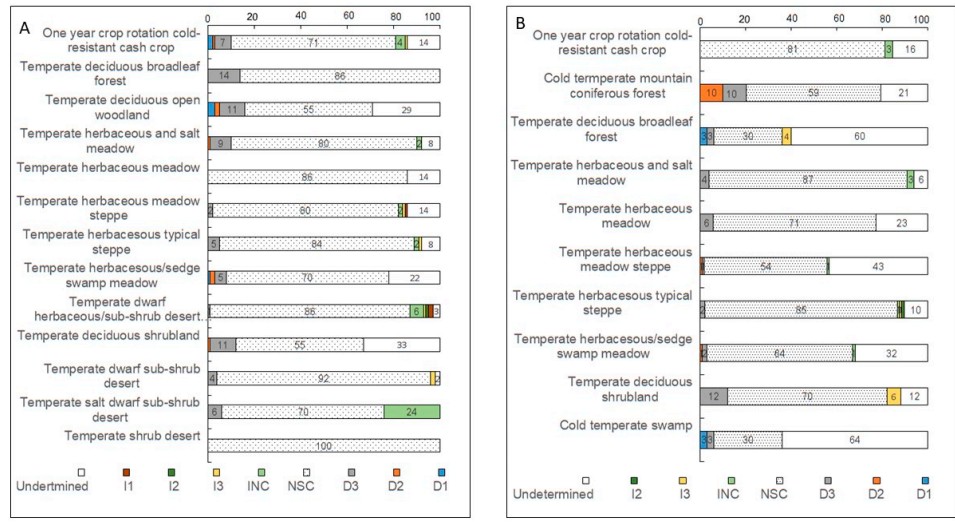

**Figure 5.** NDVI change direction (decrease/increase/no change) in different vegetation types determined by TSS-RESTREND. (**A**: Xilingol; **B**: Hulunbuir).

The years in which NDVI changed can be determined by BFAST as shown in Figure 6. In Xilingol, a change was detected after 2000, and before 2000 in Hulunbuir. These years were further confirmed by the Chow-test and some of these breakpoints were canceled due to no significant structural change to the ecosystem when double checked in the Chow-test.

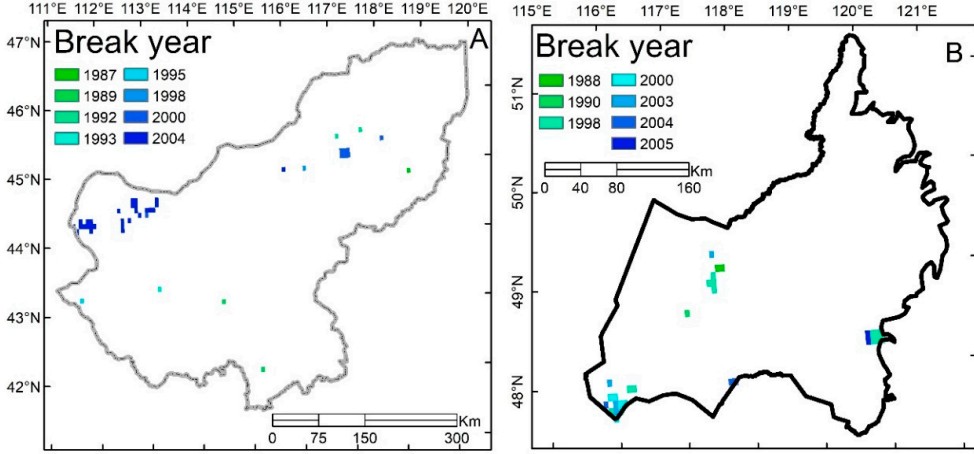

**Figure 6.** Breakpoints of NDVI detected by BFAST (**A**: Xilingol; **B**: Hulunbuir).

## 3.3. Method Comparison

We evaluated the effectiveness of the TSS-RESTREND method by plotting spatial differences between TSS-RESTREND and a standard RESTREND or a trend in the raw LTA (NDVIts) (see Figure 7). In Hulunbuir, there is no significant difference between TSS-RESTREND and the standard RESTREND analysis of NDVI change, but in Xilinggol, some pixels in western areas that are greening could be underestimated by the standard RESTREND.

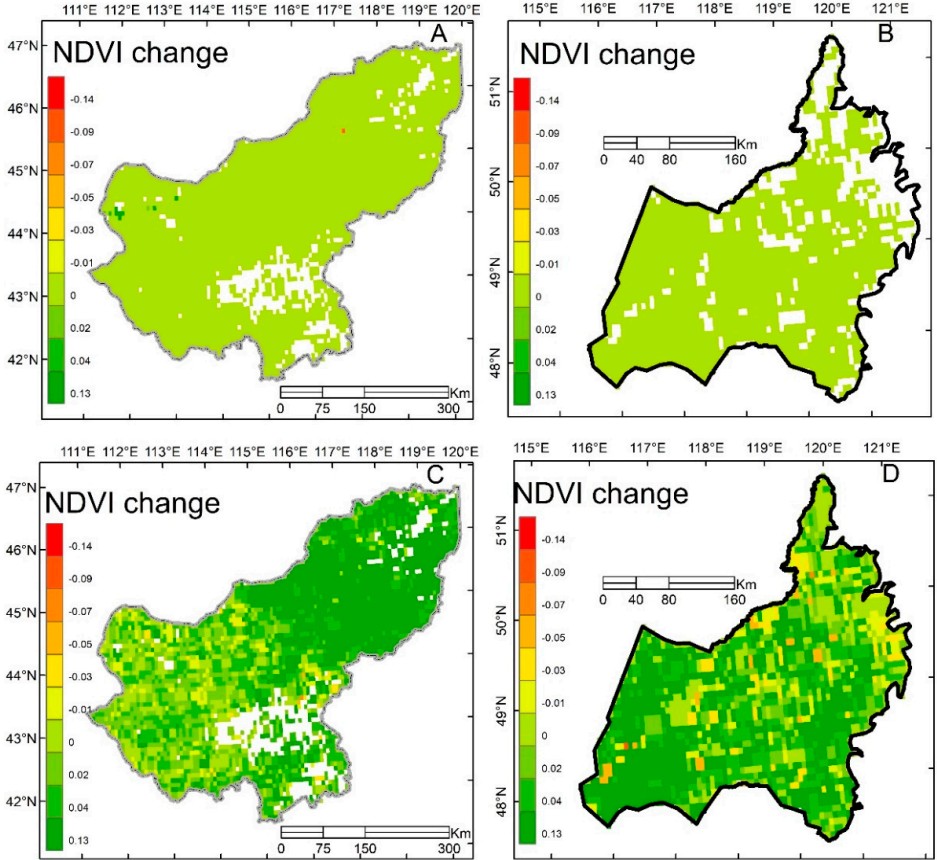

**Figure 7.** NDVI change difference between different methods in Xilingol and Hulunbuir. (**A**,**B**) The results of TSS-RESTREND minus RESTREND trend which indicates effects on break year detection on NDVI time series; (**C**,**D**) the results of TSS-RESTREND minus NDVImax trend which indicates to what extent of NDVImax trend is attributed to rainfall changes.

## 4. Discussion

### 4.1. Performance of TSS-RESTREND, RESTREND, and LTA

In areas with low interannual climatic variability, vegetation phenology is relatively stable, which means that breakpoints detected in the growth cycle using BFAST can be attributed to disturbances [18,19] or maximum NDVI regression methods like LTA can acquire abnormal deviation caused by human activities. However, with high interannual climatic variability, drought can lead to significant natural changes in phenology, which makes the separation of natural variability from environmental change complex [8,10,48]. Our results indicate that the LTA method can overestimate the land degradation in the grasslands of Xilingol and Hulunbuir. Evaluation of land degradation in this area without considering climatic variability would be incorrect [24]. Therefore, there is a need for grassland degradation studies to isolate climatic effects, especially precipitation in water-limited ecosystems, before drawing conclusions from trends in VI time series [40]. When using TSS-RESTREND, less than 1% of the pixels were selected with breakpoints detection in Xilingol (none in Hulunbuir) compared with RESTREND. These pixels indicated that there were significant structural changes in the ecosystem in the specified year. The similar results between RESTREND and TSS-RESTREND in Hulunbuir may result because NDVI changes cannot be derived from coarse remote sensing data [49]. Using finer resolution NDVI dataset may improve the accuracy of the land degradation analysis and include more spatial details. Future studies are needed to explore the optimal spatial resolution or pixel size in the application of the TSS-RESTREND analysis.

Although precipitation is considered as the main climatic driver for vegetation growth in our study area [25], partial and joint effects of precipitation and air temperature on vegetation growth could be different over precipitation zones and types of vegetation. For example, in our study area, some change of vegetation types such as temperate deciduous shrubland (33% undetermined), temperate deciduous open woodland (29% undetermined), and temperate herbaceous/sedge swamp meadow (22% undetermined) cannot be determined by TSS-RESTREND in Xilingol (see Figure 5). In addition, the TSS-RESTREND method did not work in areas of meadow steppe with temperate herbaceous meadow with 23% undetermined in Hulunbuir (see Figure 5), as found in other studies [9,40]. A possible reason may be the high soil moisture along rivers or for the forest-steppe on the slopes of mountain ranges with meltwater in spring [9,40,50]. However, this low undetermined percentage does not influence our understanding of vegetation change in the study areas.

### 4.2. Validity of Breakpoint Detection

Compared with standard RESTREND, TSS-RESTREND can detect breakpoints in ecosystem changes with vegetation and precipitation relationships. The improved detection is only valid if the breakpoints are real and not artifacts caused by noise in the dataset [8]. Burrell, Evans, and Liu [10] also analyzed and compared different vegetation dataset impacts on land degradation and found the GIMMS3g dataset caused significant errors in the trend over some of Australia's dryland regions due to sensor transition. Those problematic transition periods are from September 1994 to January 1995, November 2000 and January 2009 [8]. However, more than 95% of the breakpoints we detected are not at these times. For example, we showed a result of TSS-RESTREND in Figure 8. The breakpoint for this pixel is in the year of 1993. The total change is calculated by adding the VPR break height (BH) to the residual change (rc) with a total change for the pixel of −0.0144, which indicates land degradation. However, the standard RESTREND has no change for this pixel, which verified that when degradation occurred in the middle of the time series, RESTREND may lead to unreliable results. Based on those breakpoint results shown in Figure 8, we further checked the surface reflectance in the pixel using Landsat images in GEE, as shown in Figure 9, which is the same place as Figure 8.

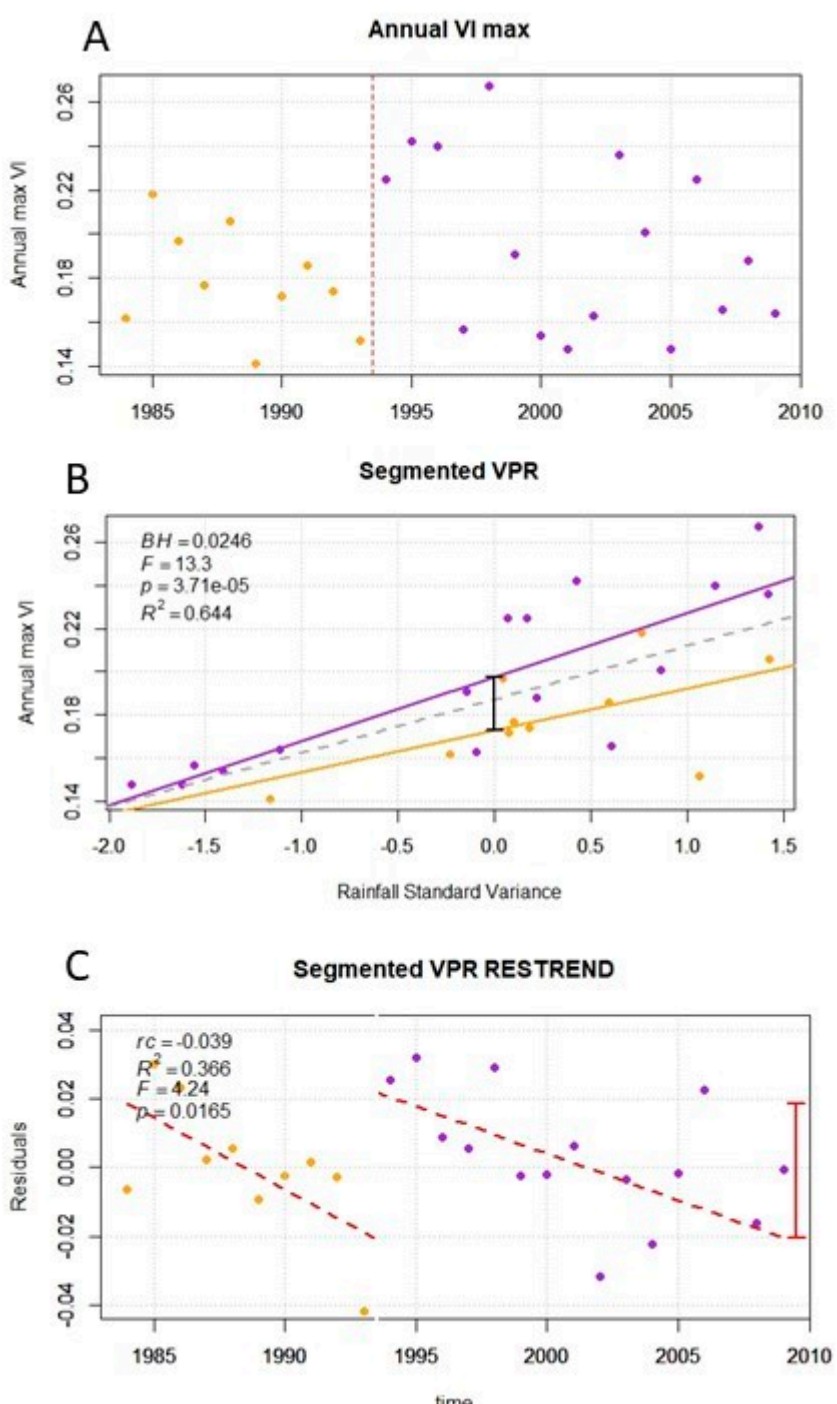

**Figure 8.** An example pixel from western Xilingol with a breakpoint in the VPR. The geographic location of the pixel is (113.189°E, 43.405°N, 113.26°E, 43.335°N). (**A**) NDVImax vs. time. The vertical red dotted line shows the position of the detected breakpoint (break year = 1993). (**B**) The change in the VPR before (orange) and after (purple) the breakpoint. The dotted grey line represents the VPR that was fitted to the data by a standard RESTREND and the black bar represents the break height (BH = 0.0246). (**C**) The segmented RESTREND applied using the segmented VPR. The red bar indicates the residual change (rc = −0.039). The total change is calculated by adding the VPR break height (BH) to the residual change (rc). So the total change is T.C. = −0.0144.

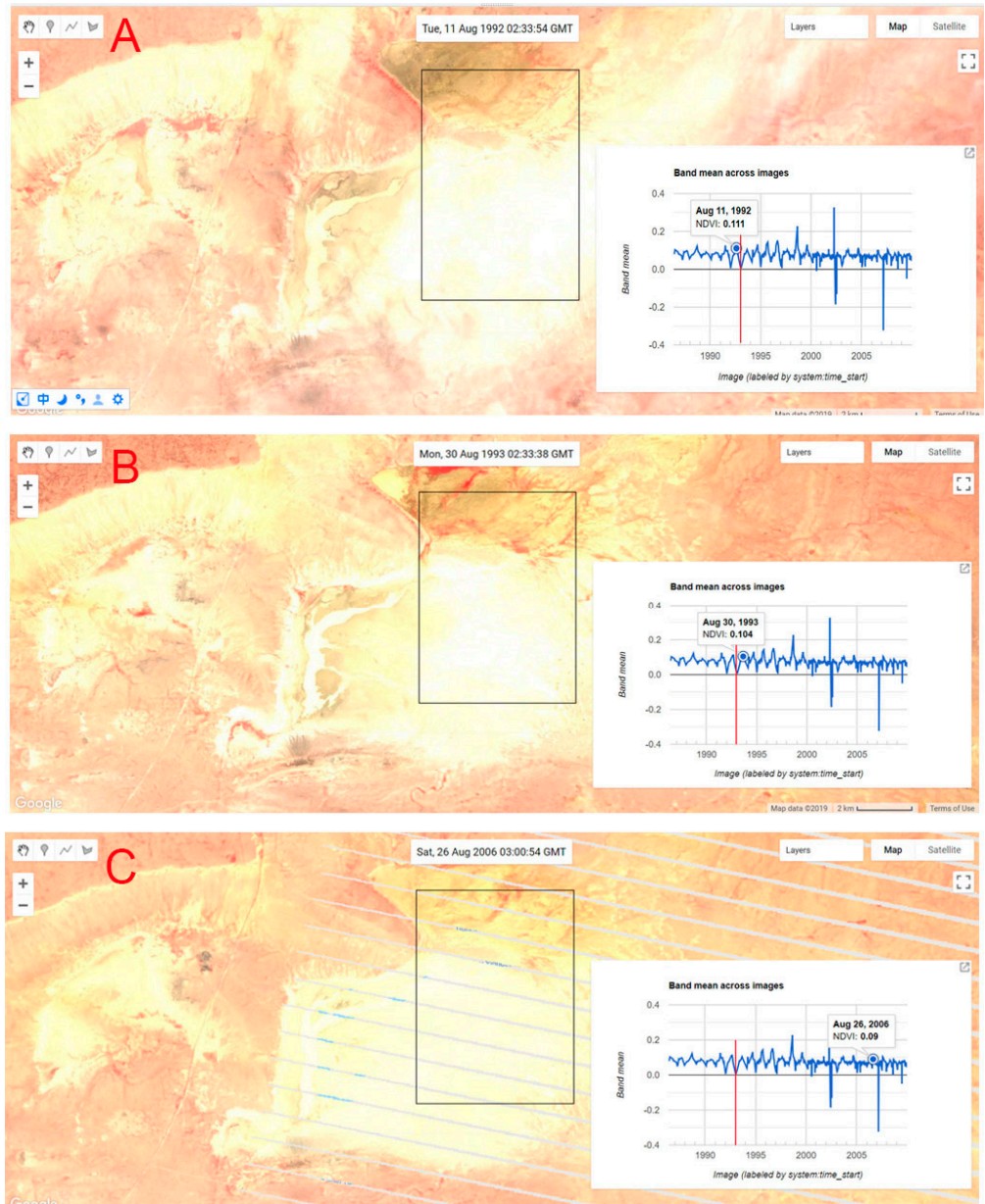

**Figure 9.** A breakpoint example validated in GEE for the same place as in Figure 8. All the images were acquired in August. The polygon boundary (black box) marks the approximate extent of the NDVI GIMMS pixel. The break year was detected by TSS-RESTREND in 1993 marked with a red line in the NDVI time series chart. The combination used to create RGB composites of Landsat images are Near Infrared, Red, and Green. The decrease of NDVI and degraded vegetation could be observed in the time series images.

Although Abel et al. [17] found that the location and timing of the breakpoints (i.e., actually simulated key moments of degradation) were more often correctly detected when applying BFAST on the SeRGS time series as compared to the RESTREND residuals, visual interpretation of Landsat optical images around breakpoints, and NDVI values verified that those breakpoints are valid in our study area. For the example presented in Figures 8 and 9, the response of VPR is changed because of successive precipitation decreases after the year of 1998, as shown in Figure 10. The rainfall shown in Figure 10 is the optimal accumulated precipitation calculated as described in Section 2.3.1.

**Figure 10.** Time series of NDVI and rainfall for the pixel in Figures 8 and 9.

However, some areas in Hulunbuir suffered from fires before 1990 [51]. The TSS-RESTERND method cannot identify those breakpoints due to relatively small areas of fire scars compared with coarse spatial resolution vegetation dataset.

### 4.3. Land Degradation

Most parts of our study areas had stable vegetation greenness in the study period, which is confirmed by other methods [52]. Although many recent studies have focused on the indicators which are used to define grassland degradation, there is still debate regarding the definition of grassland degradation at different temporal and spatial scales [53]. Remote sensing based methods can capture the greenness trend of vegetation, but we cannot be sure that degradation did not happen even if the stable greenness exists. Hulunbuir et al. [53] conducted field investigations and analyses to compare species composition between different times and their results showed that grasslands in the study area were seriously degraded. Analyses of desertification and grasslands above ground biomass (AGB) on a shorter time scale indicated that the grassland ecosystem was recovering, whereas analyses of species composition indicated that the grassland ecosystem was degrading. We also estimated AGB based on remote sensing and field data for forest management [54] and in this case the grassland AGB as an indicator of grassland degradation would be our future research direction.

Land degradation derived from the RESTREND method varies with the temporal scale due to the computation of the NDVI-rainfall regression and the residual trends. Although the BFAST method for TSS-RESTREND can detect break years throughout the study period, the results of TSS-RESTREND also depend on the temporal scale due to BFAST method on residuals.

Since 2000, ecological programs in China have been in place to reduce land degradation and relieve human pressure on land by converting cropland to grassland. Our study areas with implemented ecological programs have more vegetation gain, cropland retirement [55] or livestock grazing limitation [9]. Analytical investigations and field observations indicated increasing trends in vegetation cover and biomass production in Xilingol because of decreases in livestock production due to grassland restoration policies [56,57], in which vegetation greenness increased during the study period over typical western steppe and desert steppe in Xilingol (see Figures 2 and 3). Another possible reason for the increased greenness may result from shrub encroachment in these grasslands [58], and shrubland pixels can represent higher NDVI values than grassland pixels. For the pixels with decreased NDVI in both areas, the main driver comes from overgrazing over the study period [9]. Although other human activities, such as urbanization, must have played a significant role in developing the spatiotemporal pattern of those land degradation, where the abrupt land cover change occurred. RESTREND based method cannot be applied to these areas because the relationship between the variations in the precipitation and vegetation can collapse when those abrupt land cover changes

occurred. However, there was a decreasing trend of mining area in Xilingol and Hulunbuir thanks to the closing of many small-run businesses after 2010 [56].

## 5. Conclusions

This paper explores for eastern grasslands in China (Xilingol and Hulunbuir) the use of TSS-RESTREND, an extended version of the RESTREND methodology that considers breakpoint detection to identify pixels with abrupt ecosystem changes which violate the assumptions of a standard RESTREND.

Our analysis found that 6% and 3% of the pixels decreased in greenness between 1984 and 2009, 80% and 73% had no significant change and 5% and 3% increased. The remaining 9% and 21% were found to be indeterminate. A standard RESTREND may underestimate the greening trend in Xilingol, but both TSS-RESTREND and RESTREND analysis had no significant difference in Hulunbuir. Further work should be conducted for land degradation with TSS-RESTRREND in our study area using higher spatial resolution time-series remotely sensed images.

TSS-RESTREND has its limitations for the application of land degradation. It cannot be used in urbanized areas because the relationship between the variation in the precipitation and vegetation can collapse when this kind of land cover change occurred, which indicates that TSS-RESTREND cannot be used in abrupt change detection but gradual change. The current time series dataset covers from 1984 to 2009 and allows breakpoints detected only between 1987 and 2005. Changes occurring within the first three years, or the last three years, of the time series, cannot be detected by TSS-RESTREND, which indicates that the break detection procedure needs a relative learning period during which changes from one stable steady-state to another one occurred.

**Author Contributions:** C.L. and J.M. designed the study. C.L. and Y.T. conducted data analysis and wrote the paper. H.H. contributed to writing. J.J. contributed to writing. X.F. contributed to collecting data. Z.Z. helped to validations.

**Funding:** This work was supported by The National Key Research and Development Program of China (2016YFC0501101 and 2016YFC0503603) and Open Fund of State Key Laboratory of Remote Sensing Science (OFSLRSS201704). The China Scholarship Council funded the first author studying abroad.

**Acknowledgments:** We thank Burrell Arden at Climate Change Research Centre in the University of New South Wales Sydney for discussions of TSS-RESTREND and use of different versions of GIMMS NDVI datasets, Yuanyuan Zhao at the University of California, Berkeley for production of historic precipitation raster images and Quansheng Li at State Key Laboratory of Water Resource Protection and Utilization in Coal Mining for prior knowledge sharing and discussion in the study areas during several seminars and conferences. We thank the three reviewers and the guest editor for their constructive suggestions.

**Conflicts of Interest:** The authors declare no conflict of interest.

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
