# Peer review of "Detecting Land Degradation in Eastern China Grasslands with Time Series Segmentation and Residual Trend analysis (TSS-RESTREND) and GIMMS NDVI3g Data"

_remotesensing, doi:10.3390/rs11091014_

Round 1

Reviewer 1 Report

Answer Question 1 is answer on slightly different question, because in my question I wondered about another level – more physical level of soil moisture models.

Answer on Question 2 is OK.

Answer on Question 3 is slightly strange and not fully understandable. I propose to read it again and to try to improve the main sense. Answer should be improved.

Answer on Question 4. Is it enough to use only “monthly accumulation precipitation dataset”? (Question is still open, because rain can be nonuniform and some interpolations and accumulated precipitation within the 1 month could not reflect this irregularity, which can be reason for droughts. Answer should be refined.

Answer on Question 5. Information on Credentials usage should be explicitly mentioned. Authors should remove this part form the paper or provide procedure for mentioned here link’ usage.

Answer on Question 6. Answer should be refined. One model should provide similar results (one input – one output) or authors should provide model usage conditions, limitations, examples of model verification in the manuscript

Answer on Question 7. Is it enough to mention validation images only? For photointerpretation only? This is not enough here.

Author Response

Answer Question 1 is answer on slightly different question, because in my question I wondered about another level – more physical level of soil moisture models.

Response: Data on soil moisture are not available for the region or at the spatial scale of our study. However, it is possible the soil moisture could influence the relation between vegetation and precipitation, and we have added some text on this.         

Although precipitation is considered as the main climatic driver for vegetation growth in our study area [24], partial and joint effects of precipitation and air temperature on vegetation growth could be different over precipitation zones and types of vegetation. For example, in our study area, some change of vegetation types such as temperate deciduous shrubland (33% undetermined), temperate deciduous open woodland (29% undetermined) and temperate herbaceous/sedge swamp meadow (22% undetermined) cannot be determined by TSS-RESTREND in Xilingol (see Figure 5). In addition, the TSS-RESTREND method did not work in areas of meadow steppe with temperate herbaceous meadow with 23% undetermined in Hulunbuir (see Figure 5), as found in other studies [9,39]. A possible reason may be the high soil moisture along rivers or for the forest steppe on the slopes of mountain ranges with meltwater in spring [9,39,49]. However, this low undetermined percentage does not influence our understanding of vegetation change in the study areas.

  Answer on Question 2 is OK.

Answer on Question 3 is slightly strange and not fully understandable. I propose to read it again and to try to improve the main sense. Answer should be improved.

Response: We first answer the question, ‘What is the model of VPR?’. VPR is the abbreviation for vegetation-precipitation relationship. For each pixel, we consider that there is a linear relationship between vegetation and precipitation. An Ordinary Least Squares Regression (OLSR) is calculated between peak growing season NDVI (NDVImax) and the optimal accumulated precipitation. Using the VPR model, we get the predicted NDVI value for each pixel according to the optimal accumulated precipitation.

Secondly, why breakpoints are happened? In the standard RESTREND method, breakpoints are not detected, which means that for a pixel, the VPR linear model is executed during the whole span of the study period. In some cases, this linear relationship can be broken in the middle of the time series and detecting breakpoints have been verified as necessary to improve drylands.

Thirdly, ‘Maybe it makes sense to use some additional model-based (non-linear) dependencies in such a case?’ Obeying the linear relationship is our hypothesis with using RESTREND and TSS-RESTREND. In this paper, we do not discuss the relationship pattern between vegetation and precipitation for each pixel.

Answer on Question 4. Is it enough to use only “monthly accumulation precipitation dataset”? (Question is still open, because rain can be nonuniform and some interpolations and accumulated precipitation within the 1 month could not reflect this irregularity, which can be reason for droughts. Answer should be refined.

Response: Monthly precipitation dataset is the only recent dataset we could get from the National Meteorological Information Center of China. It was calculated based on daily observations. In our study, we calculated optimal accumulation of precipitation pixel by pixel to determine the combination of accumulation period (1-12 months) and offset period (1-3 months) that produced the highest correlation coefficient with NDVImax. In detail, we calculated the possible accumulated precipitation table (ACP) from 1984-2009.

Answer on Question 5. Information on Credentials usage should be explicitly mentioned. Authors should remove this part form the paper or provide procedure for mentioned here link’ usage.

Response: We think it’s better to keep the code address in the manuscript to benefit our community. People who what to use the codes could get it, though registration is necessary. To do so click the website https://earthengine.google.com/ and hit the upper right ‘Sign up’ to get an account.

Answer on Question 6Answer should be refined. One model should provide similar results (one input – one output) or authors should provide model usage conditions, limitations, examples of model verification in the manuscript

Based on our field work, Hulunbuir grassland has better condition and protection than Xilingol, which can be observed in Figures 3 and 4. The main reason for the results in Hulunbuir may be that the VPR did not break down during the study period. We added information on the method’s limitation in conclusion.

TSS-RESTREND has its limitations for the application to land degradation. It cannot be used in urbanized areas because the relationship between the variation in the precipitation and vegetation can collapse when this kind of land cover change occurred, which indicates that TSS-RESTREND cannot be used in abrupt change detection but gradual change. The current time series dataset covers from 1984 to 2009 and allows breakpoints detected only between 1987 and 2005. Changes occurring within the first 3 years, or last 3 years, of the time series cannot be detected by TSS-RESTREND, which indicates that the break detection procedure needs a relative learning period to allow changes from one stable steady-state to another one.

Answer on Question 7. Is it enough to mention validation images only? For photointerpretation only? This is not enough here.

Response: Given the coarse spatial resolution of GIMMS3g data, field work is difficult and the few field photos are not sufficient for the validation. Landsat time series data can be used, but depend on careful matching of dates and of different spatial scales. In the Google Earth Engine , thousands of Landsat images are available.For example in Figure 10 to compare vegetation conditions in the growing season in August, the NDVI time series for the day of interest is selected.  Good vegetation conditions (red) can be observed on the top of the pixel in 1992 and 1993, but poor vegetation conditions (vanishing red) are shown in the year of 2006.

Reviewer 2 Report

The authors have performed further analyses to address my concerns. Adding Figs. 8-10 make this work more solid. I have no further comments.

Author Response

Comments and Suggestions for Authors

The authors have performed further analyses to address my concerns. Adding Figs. 8-10 make this work more solid. I have no further comments.

Reviewer 3 Report

The revised version of the manuscript is improved. However, three areas still need some work.

First, the new figure 9 needs revising, as it's not very clear what it's showing. It would be best to remove D,E,F as they are not helpful. The polygon should have a boundary but be transparent, as it's difficult to see what's going on inside the polygon. The chart could be increased in size and shown once with A, B, C, D etc on the chart to show the timings on the different images. The chart could also have red line to mark the detected breakpoint timing. The title for the figure should say that the polygon boundary marks the approximate extent of the NDVI GIMMS pixel. The are shown should be cropped more tightly to the polygon/pixel area. The figure should also explain what combination of landsat bands are being shown. The figure probably just needs the images for1992 (as the before image), 1993 (as the change image) and 2006 (as the after image).

Second, I don't understand Figure 10, as the text suggests that the breakpoint in 1993 (figure 9) is due to changes in precipitation after 1998. Also, it's does look 1998 was a dry year, but 1989 looks quite wet. I don't know whether having the total annual precipitation might be clearer here, rather than monthly or whatever is currently plotted in Figure 10.

Third, I think the conclusions should provide a summary of when TSS-RESTREND is applicable i.e. need a link between veg and precipitation (so not suitable for urban areas, or areas that change to urban); the current time-series covers 1984-2009 and allows breakpoints between 1984-2005 to be detected, changes occurring within the first 3 years, or last 3 years, of the time-series will not be detected by TSS-RESTREND etc. What type of change is appropriate for (abrupt change/gradual change). Also need to make the point that the change needs to be from one consistent state to another consistent state.

Author Response

The revised version of the manuscript is improved. However, three areas still need some work.

First, the new figure 9 needs revising, as it's not very clear what it's showing. It would be best to remove D,E,F as they are not helpful. The polygon should have a boundary but be transparent, as it's difficult to see what's going on inside the polygon. The chart could be increased in size and shown once with A, B, C, D etc on the chart to show the timings on the different images. The chart could also have red line to mark the detected breakpoint timing. The title for the figure should say that the polygon boundary marks the approximate extent of the NDVI GIMMS pixel. The are shown should be cropped more tightly to the polygon/pixel area. The figure should also explain what combination of landsat bands are being shown. The figure probably just needs the images for 1992 (as the before image), 1993 (as the change image) and 2006 (as the after image).

Response: We followed most of your suggestions and modified the figure. We did not plot the NDVI time series in a separate frame because keeping both in the same frame could be benefit visual interpretation easily online (Google earth engine) at the same time.

Figure 9. A breakpoint example validated in GEE for the same place as in the Figure 8. All the images were acquired in August. The polygon boundary (black box) marks the approximate extent of the NDVI GIMMS pixel. The break year was detected by TSS-RESTREND in 1993 marked with red line in the NDVI time series chart. The combination used to create RGB composites of Landsat images are Near Infrared, Red and Green. The decrease of NDVI and degraded vegetation could be observed in the time series images.

Second, I don't understand Figure 10, as the text suggests that the breakpoint in 1993 (figure 9) is due to changes in precipitation after 1998. Also, it's does look 1998 was a dry year, but 1989 looks quite wet. I don't know whether having the total annual precipitation might be clearer here, rather than monthly or whatever is currently plotted in Figure 10.

Response: The precipitation in Figure 10 is accumulated precipitation. As shown in Section 2.3.1, the optimal accumulation of precipitation is calculated pixel by pixel to determine the combination of accumulation period (1-12 months) and offset period (1-3 months) that produced the highest correlation coefficient with NDVImax.

We added extra sentences for explaining this in the manuscript.

Visual interpretation of Landsat optical images around breakpoints and NDVI values verified that those breakpoints are valid. For the example presented in Figure 8 and 9, the response of VPR is changed because of successive precipitation decreases after the year of 1998 as shown in Figure 10. The rainfall shown in the Figure 10 is the optimal accumulated precipitation calculated as described in Section 2.3.1.

Third, I think the conclusions should provide a summary of when TSS-RESTREND is applicable i.e. need a link between veg and precipitation (so not suitable for urban areas, or areas that change to urban); the current time-series covers 1984-2009 and allows breakpoints between 1984-2005 to be detected, changes occurring within the first 3 years, or last 3 years, of the time-series will not be detected by TSS-RESTREND etc. What type of change is appropriate for (abrupt change/gradual change). Also need to make the point that the change needs to be from one consistent state to another consistent state.

Response: We added in the conclusion.

TSS-RESTREND has its limitations for the application of land degradation. It cannot be used in urbanized areas because the relationship between the variation in the precipitation and vegetation can collapse when this kind of land cover change occurred, which indicates that TSS-RESTREND cannot be used in abrupt change detection but gradual change. The current time series dataset covers from 1984 to 2009 and allows breakpoints detected only between 1987 and 2005. Changes occurring within the first 3 years, or last 3 years, of the time series can not be detected by TSS-RESTREND, which indicates that the break detection procedure needs a relative learning period in which let changes from one stable steady-state to another one.

Acknowledgments: We thank Mr. Burrell Arden at Climate Change Research Centre in the University of New South Wales Sydney for discussions of TSS-RESTREND and use of different versions of GIMMS NDVI datasets, PhD. Yuanyuan Zhao at University of California, Berkeley for production of historic precipitation raster images and PhD. Quansheng Li at State Key Laboratory of Water Resource Protection and Utilization in Coal Mining for prior knowledge sharing and discussion in the study areas during several seminars and conferences. Great gratitude should be delivered to the three reviewers and the guest editor for their constructive suggestions.  

This manuscript is a resubmission of an earlier submission. The following is a list of the peer review reports and author responses from that submission.

Round 1

Reviewer 1 Report

Some open questions.

It’s not clear, why “strong linear relationship between the variations in the precipitation and vegetation index” is so popular. It’s natural, if this relation is taken into account. But this dependency is very complex and cannot be just linear. Do authors have information on other approaches for that? How to catch “moisture reserve” in this case? Using soil structure, for example.

2. When degradation occurred in the middle of time series, a strong linear relationship may be lacking and lead to unreliable results. Is it usual situation?

3. Why breakpoints are happened? What is the model of VPR? Maybe it makes sense to use some additional model-based (non-linear) dependencies in such a case?

4. Which meteorological data are used by authors? With which structure and resolution?

5. Link http://cdcNaNa.gov.cn/home.do (meteorological data) is unavailable, for access to GEE with code I need registration credentials. O don’t have ones.

6. A standard RESTREND may underestimate the greening trend 379 in Xilingol, but both TSS-RESTREND and RESTREND analysis had no significant difference in 380 Hulunbuir. Is it means that results of both algorithms are depending on territory of interest and can provide different results? How authors can formulate the generic principle of such a method applicability? And what authors can say about in-situ validation of the methods?

7. Do authors use any in-situ up-to-date information for, for example, brealpoint's collection?

Author Response

1. It’s not clear, why “strong linear relationship between the variations in the precipitation and vegetation index” is so popular. It’s natural, if this relation is taken into account. But this dependency is very complex and cannot be just linear. Do authors have information on other approaches for that? How to catch “moisture reserve” in this case? Using soil structure, for example.

Response: It’s natural that this linear relationship is frequently used in arid area. There are two main methods to distinguish human-induced land degradation in arid area from inter-annual variability in rainfall. The first is Rain Use Efficiency, RUE (RUE=net primary production (NPP)/Rainfall or Normalized Difference Vegetation Index (NDVI)/Rainfall). Another method is Residual Trends (RESTREND). RUE had a very strong negative correlation with rainfall and varied between years (Wessels et al. 2007) and RESTREND method has been recommended (Li et al. 2012, Wessels et al. 2007). A recent work (Kundu et al. 2017) also verified that RUE is not too useful in the areas with climate and human-induced desertification processes, whereas RESTREND could able to identify the areas with human-induced desertification.

As far as we know, there is one study that considering soil moisture when conducting RESTREND in western Africa (Ibrahim et al. 2015) and it shows that the soil moisture/NDVI pixel-wise residual trend indicates degraded areas more clearly than rainfall/NDVI.

Wessels, Konrad J., et al. "Can human-induced land degradation be distinguished from the effects of rainfall variability? A case study in South Africa." Journal of Arid Environments 68.2 (2007): 271-297.           

Li, Ang, Jianguo Wu, and Jianhui Huang. "Distinguishing between human-induced and climate-driven vegetation changes: a critical application of RESTREND in inner Mongolia." Landscape ecology 27.7 (2012): 969-982.             

Kundu, Arnab, et al. "Desertification in western Rajasthan (India): an assessment using remote sensing derived rain-use efficiency and residual trend methods." Natural Hazards 86.1 (2017): 297-313.

Ibrahim, Yahaya, et al. "Land degradation assessment using residual trend analysis of GIMMS NDVI3g, soil moisture and rainfall in Sub-Saharan West Africa from 1982 to 2012." Remote Sensing 7.5 (2015): 5471-5494.

2. When degradation occurred in the middle of time series, a strong linear relationship may be lacking and lead to unreliable results. Is it usual situation?

Respond: This situation could be observed in a simulation study by Wessels et al. 2012 that a degradation of greater that 20% occurring in the middle of the time series will cause an otherwise strong linear relationship to breakdown, making RESTREND results unreliable. When degradation occurs within the time series, the strength of this relationship is reduced and the NDVI predicted from the rainfall by a linear model becomes less reliable. They demonstrated that 20% degradation in the middle of the time series leads to low R2 values of 0.3 to 0.4, or as low as 0.2. At a 30% degradation intensity, the R2 values in most of the simulation combinations were below 0.3–0.4. Under these circumstances the regression becomes an unreliable predictor of NDVI based on the rainfall, which implies that any trends observed in the RESTREND residuals beyond 20% intensity may not be caused by the simulated degradation, but rather by outliers due to decorrelation between rainfall and NDVI.

Wessels, Konrad J., F. Van Den Bergh, and R. J. Scholes. "Limits to detectability of land degradation by trend analysis of vegetation index data." Remote sensing of Environment 125 (2012): 10-22.

3. Why breakpoints are happened? What is the model of VPR? Maybe it makes sense to use some additional model-based (non-linear) dependencies in such a case?

Response: The VPR is the vegetation-precipitation relationship. An Ordinary Least Squares Regression (OLSR) is calculated between peak growing season NDVI (NDVImax), a proxy for NPP [7,8], and the optimal accumulated precipitation. Breakpoints happened when degradation occurred in the middle of time series, a strong linear relationship may be lacking and lead to unreliable results. In the introduction section, we added information starting at Line 66.

applied to land cover mapping [12-14] and forest management [15,16].It has been argued thatthe dryland degradation could be improved when detecting breakpoint when the relationship between vegetation and precipitation breaksdown [5,7].

4. Which meteorological data are used by authors? With which structure and resolution?

Response: We use the monthly accumulation precipitation dataset, produced with the thin-plate spline spatial interpolation using meteorological station data.

5. Link http://cdcNaNa.gov.cn/home.do (meteorological data) is unavailable, for access to GEE with code I need registration credentials. O don’t have ones.

Response: Both of the websites require registration.

6. “A standard RESTREND may underestimate the greening trend 379 in Xilingol, but both TSS-RESTREND and RESTREND analysis had no significant difference in 380 Hulunbuir.” Is it means that results of both algorithms are depending on territory of interest and can provide different results? How authors can formulate the generic principle of such a method applicability? And what authors can say about in-situ validation of the methods?

Response: TSS-RESTREND method needs in-situ validation, but it is difficult with the use of the coarse spatial resolution dataset. Based on our knowledge, Hulunbuir grassland has better condition and protection than Xilingol, which can be observed in Figures 3 and 4. The main reason for the results in Hulunbuir may be that the VPR did not break down during the study period.

7. Do authors use any in-situ up-to-date information for, for example, brealpoint's collection?

Response: We added validation images in the Discussion section.

detectionValidity of breakpoint detection

Compared with standard RESTREND, TSS-RESTREND can detect breakpoints in ecosystem changes with vegetation and precipitation relationships. The improved detection is only valid if the breakpoints are real and not artefacts caused by noise in the dataset [8]. Burrell, Evans and Liu [10] also analyzed and compared different vegetation dataset impacts on land degradation and found the GIMMS3g dataset caused significant errors in the trend over some of Australia’s dryland regions due to sensor transition. Those problematic transition periods are from September 1994 to January 1995, November 2000 and January 2009 [8]. However, more than 95% of the breakpoints we detected are not at these times. For example, we showed a result of TSS-RESTREND in Figure 8.The breakpoint for this pixel is the year of 1993. The total change is calculated by adding the VPR break height (BH) to the residual change (rc)withatotal changefor the pixelof-0.0144,which indicates land degradation.However, the standard RESTRENDhasno change for this pixel, which verified thatwhen degradation occurred in the middle of time series,RESTRENDmay lead to unreliable results.Based on those breakpoint results shown in Figure 8,we furtherchecked the surface reflectance in the pixel using Landsat imagesin GEE, as shown in Figure 9, which is the same placeasFigure 8.

Figure 8.An example pixel from western Xilingol with a breakpointin the VPR. The geographic location of the pixelis(113.189°E,43.405°N,113.26°E,43.335°N).A)NDVImax vs. time.The vertical red dotted lineshows the position of the detected breakpoint(break year = 1993). B) The change in the VPR before (orange) and after (purple)the breakpoint. The dotted grey line represents the VPR thatwas fitted to the data by a standard RESTREND and the black bar represents the break height (BH=0.0246).C) The segmented RESTREND applied using the segmented VPR. The red bar indicates the residual change (rc = -0.039). The total change is calculated by adding the VPR break height (BH) to the residual change (rc). So the total change is T.C.= -0.0144.

Figure 9.Abreakpointexamplevalidated in GEEfor thesame placeasin the Figure 8.All the images were acquired inAugust.Unvalid Landsat NDVI images should be excluded (D,E,F).The breakyear was detected by TSS-RESTREND in 1993.The decrease of NDVIand degradedvegetation could be observed.

Visual interpretation of Landsat optical images around breakpoints and NDVI values verified that those breakpoints are validand represented structural changes.For the example presented in Figure 8 and 9,the response of VPR is changed because ofsuccessive precipitation decreasesafter the year of1998as shown in Figure 10.

Figure 10.Time series of NDVI and rainfall for the pixel in Figure 8 and 9.

However,Butthere is one case that TSS-RESTREND method cannot detect in the study area with GIMMS dataset is fire scars.  sSome areas in Hulunbuir suffered from fires before 1990 [5048]. TSS-RESTERND method cannot identify those breakpoints due to relatively small areas of fire scars compared with coarse spatial resolution vegetation dataset.

Reviewer 2 Report

I have read this paper with interests. The authors proposed to use Time Series Segmentation and Residual Trend analysis (TSS-RESTREND) to judge whether vegetation growth condition has changed or not, and compared it with two other methods, which yielded different change trends. I believe the contents of this paper is suitable for Remote Sensing, and the comparison of three methods can be helpful for other researchers. I therefore suggest the publication of this research.

However, I do think the paper needs to be revised before it can be accepted. I felt confused several times when reading it:

1.      The introduction emphasizes on distinguishing human impacts from climate changes in controlling vegetation changes. But how did the comparison contribute to this topic?  The methods are able to detect changes, but actually have nothing to do with drivers of changes.

2.      Another issue is: the authors compared three method and assumed implicitly that TSS-RESTREND is the best and should be used as a standard. How can this assumption be validated? This is the most critical issue. Without a proper demonstration, comparisons can be carried out on numerous methods, but we still can’t draw a conclusion on which one we should trust. The authors should give more evidences or explanation, either from their own or other researches, on the trustability of TSS-RESTREND. The best solution would be to acquire ground monitoring data, but can be very difficult.

3.      The authors mentioned (section 2.2.4) that they used Landsat time-series data, but I didn’t see clearly in the result section how and for which purpose the landsat images were used (only several sentences Lines 334-336 without further figures or tables). I was expecting the authors to use the Landsat images to show which part of the grasslands were converted to other land-use types, such as coal-mined areas, city lands or farmlands.  If this part can be strengthened, issue 1 I raised above can be solved to some extents.

4.      Since this research was talking about grasslands, I suggested to delete the contents related with other vegetation types. Figure 5 didn’t add points to your research. Some of the related text can also be deleted.

5.      Why TSS- RESTREND and RESTREND didn’t show difference in one of your research areas? What’s the message the authors want to convey to the potential readers? Did this mean the two methods are equally good? Or should we expect some differences at all? This adds to my concern raised in issue 2. The conclusion section mentioned this difference once again, but concluded in a rush without further explanations.

I felt that most of these issues can be addressed with a clearly restructuring of your writing. Add more explanations for some issues, refine your stated research purposes…..

Some other suggestions:

Line 154,  add references for the use of max NDVI as NPP?

Some of section 2.1 can be integrated into introduction, to show why the two grasslands are important. For mining and grasslands, some potential references can be: Tao et al. 2015 PNAS 112 (7) 2281-2286 , and Wu et al. 2015 Landscape Ecol 30:1579–1598.

What’s difference between Fig.3ab and Fig.4?

Typing errors should be carefully checked, I found some, but there are more.

Line 333, add “and” at the end of this line

Line 305, “land” not “and”

Line 182, change unlike with to unlike

Author Response

1.      The introduction emphasizes on distinguishing human impacts from climate changes in controlling vegetation changes. But how did the comparison contribute to this topic?  The methods are able to detect changes, but actually have nothing to do with drivers of changes.

Response: To clarify we added.

In these areas, annul precipitation is low and inter-annual variability in rainfall high [3,4].When conductingvegetation greennesschange andland degradationanalysisin these areascaused by human activities,it isthe first steptoexclude the impactfromclimaticvarieties.The climate …

2.      Another issue is: the authors compared three method and assumed implicitly that TSS-RESTREND is the best and should be used as a standard. How can this assumption be validated? This is the most critical issue. Without a proper demonstration, comparisons can be carried out on numerous methods, but we still can’t draw a conclusion on which one we should trust. The authors should give more evidences or explanation, either from their own or other researches, on the trustability of TSS-RESTREND. The best solution would be to acquire ground monitoring data, but can be very difficult.

Response: We added.

applied to land cover mapping [12-14] and forest management [15,16].It has been argued thatthe dryland degradation could be improved when detecting breakpoint when the relationship between vegetation and precipitation break down [5,7]….

In the discussion, we expanded our analysis and validation using time series Landsat images and confirm the necessity of breakpoint detection. In the section of 4.2, we added these sentences.

For example, we showed a result of TSS-RESTREND in Figure 8.The breakpoint for this pixel is the year of 1993. The total change is calculated by adding the VPR break height (BH) to the residual change (rc). So the total changefor the pixelis -0.0144,which indicates land degradation.However, the standard RESTRENDshows there was no change for this pixel, which verified thatwhen degradation occurred in the middle of time series,RESTRENDmay lead to unreliable results.Based on those breakpoint results shown in Figure 8,we furtherchecked the surface reflectance in the pixel using Landsat imagesin GEE. As shown in Figure 9, which is the same placewithFigure 8.

3.      The authors mentioned (section 2.2.4) that they used Landsat time-series data, but I didn’t see clearly in the result section how and for which purpose the landsat images were used (only several sentences Lines 334-336 without further figures or tables). I was expecting the authors to use the Landsat images to show which part of the grasslands were converted to other land-use types, such as coal-mined areas, city lands or farmlands.  If this part can be strengthened, issue 1 I raised above can be solved to some extents.

Response: We added extra information in the Discussion as noted in response to Reviewer #1.

4.      Since this research was talking about grasslands, I suggested to delete the contents related with other vegetation types. Figure 5 didn’t add points to your research. Some of the related text can also be deleted.

Response: We inclined to keep this part for the need of knowing situations of grassland ecosystem in the study area. We want to know which vegetation categories are sensitive to TSS-RESTRAND and which categories cannot be determined, which can benefit us for preparing the next step study.

5.      Why TSS- RESTREND and RESTREND didn’t show difference in one of your research areas? What’s the message the authors want to convey to the potential readers? Did this mean the two methods are equally good? Or should we expect some differences at all? This adds to my concern raised in issue 2. The conclusion section mentioned this difference once again, but concluded in a rush without further explanations.

Response: Yes, in the Hulunbuir, TSS-RESTREND did not show superiority over RESTREND. This is because the vegetation-precipitation relationship did not break down in the Hulunbuir over the study period based on GIMMS NDVI3g dataset. We guess the possible reason might be this as writing in the section 4.1 .

Some other suggestions:

Line 154,  add references for the use of max NDVI as NPP?

Response: Yes, we added.

Some of section 2.1 can be integrated into introduction, to show why the two grasslands are important. For mining and grasslands, some potential references can be: Tao et al. 2015 PNAS 112 (7) 2281-2286 , and Wu et al. 2015 Landscape Ecol 30:1579–1598.

 Response: Yes, we added. Thanks for your suggestion.

What’s difference between Fig.3ab and Fig.4?

 Response: Fig.3 ab show the total change calculated from TSS-RESTREND, which could be referenced in Figure 8. Total change is the sum of break height (BH) and residual change (rc). Fig. 4 show the change level which illustrated in table 1.

Typing errors should be carefully checked, I found some, but there are more.

Line 333, add “and” at the end of this line

Response:

Do you mean this sentence ‘to sensor transition. Those problematic transition periods are from September 1994 to January 1995, (and?) November 2000 and January 2009 [8].

We think our original writing is right.

Line 305, “land” not “and”

Response: thanks for your finding. We’ve changed.

Evaluation ofland degradation in

Line 182, change unlike with to unlike

Response: changed.

 Unlike with the previous studies with BFAST [15,431],

Reviewer 3 Report

The manuscript takes an existing method (TSS-RESTREND) (Burrell et al., 2017) that had been applied successfully to Australia and applies it to 2 predominantly grassland areas of eastern China. The paper then compares the results of TSS_RESTREND method with RESTREND, as also done in Burrell et al., (2017).

The main problem with the paper is the lack of validation. I know that validating data sets at this scale is difficult and there is often a lack of data, but as a reader I find it very difficult to judge how well the method is working. The issues that ideally require some validation are:

1) is the change real change

2) is the timing of the breakpoint appropriate

3) are their types of change that are not detected (the paper suggests that change in non-grassland habitats is poorly detected), as are fire scars due to the small scale compared to the coarse scale of the satellite data

The paper provides some information in the discussion of these issues, but I think this section needs some expansion and some evidence. I think one of the sections in the discussion should be a ‘Validation of the detected change’ (or ‘Corroboration of the detected change’). Section 4.2 could be renamed ‘Validity of breakpoint detection’ and should include a before and after image of at least one of the changes, as well as a graph showing the breakpoint in the NDVI time series (like Figure 5 in Burrell et al 2017) so readers can understand what level of change the method is able to detect.

Minor issues:

Line 21 – 24: very long sentence – consider splitting into two sentences.

Line 34: km2, 2 should be superscript

Line 35: change ‘has’ to ‘have’

Line 50-51: slightly confusing as currently written, change phrasing from ‘maximum NDVI, a proxy for ecosystem productivity, and vegetation greenness and precipitation’ to ‘maximum NDVI, a proxy for ecosystem productivity, and precipitation’

Line 71: change ‘objective’ to aim or purpose. This separates the overall aim of the work, from the specific objectives mentioned in line 74.

Line 88: could add ‘as well as pollution from the power plants’

Line 119: should this be ‘GIMMS NDVI3g’

Line 125-126: delete ‘spatial distribution’ so text becomes ‘precipitation maps’

Line 131: change ‘from’ to ‘by’

Line 140: Change title to something like ‘Landsat processing in Google Earth Engine’ or ‘Validating breakpoint timing in Google Earth Engine’

Line 146: including the GEE code is good, but it would be nice if the code centred on the Region of Interest

Line 154: change ‘of’ to ‘for’

Line 200: change ‘may indicate that there might have been significant structure change’ to ‘may indicate a significant structural change’

Line 218: ‘Pixels where no significant model can be fitted are classified as indeterminate’

Line 220: I think only formula 3 has been corrected for a small typo in Burrell et al., 2017, so the text should make this clear something like  ‘corrected a small typo in Burrell et al., 2017 that affected equation 3’.

Line 226: I can’t see a definition for ‘NDVIts’ please add.

Line 259: change to ‘LTA, whereas TSS-RESTREND suggested degraded areas were distributed in the central and southwestern areas.’

Page 12: sections 4.2 and 4.3 have the same heading.

Author Response

Response: We expanded validation in the discussion part. The time series Landsat images (Figure 9) showed the real change of degraded trend for the corresponding pixel (Figure 8) and the possible reason may be the VPR break down for the decreasing precipitation as shown in Figure 10.

See response to Reviewer #1 for complete text added.

Minor issues:

Line 21 – 24: very long sentence – consider splitting into two sentences.

Response: Sentence was split and condensed.

Time Series Segmentation and Residual Trend analysis (TSS-RESTREND), was applied to grasslands in eastern China.TSS-RESTREND isan extended version of Residual Trend (RESTREND) methodology., that Itconsiders breakpoint detection to identify pixels with abrupt ecosystem changes which violate the assumptions of RESTREND., was applied to grasslands in eastern China.With TSS-RESTREND,Iin Xilingol (111°59′-120°00′E and 42°32′-46°41′E) and Hulunbuir (115°30′-122° E and 47°10′-51°23′N)grassland,

Line 34: km2, 2 should be superscript

Line 35: change ‘has’ to ‘have’

Response: Two changes made.

Grasslands are the largest ecosystem in China, covering 393,000 km2, 43% of the country territory [1]. In the past forty decades, the grasslands in China haves experienced serious land degradation…

Line 50-51: slightly confusing as currently written, change phrasing from ‘maximum NDVI, a proxy for ecosystem productivity, and vegetation greenness and precipitation’ to ‘maximum NDVI, a proxy for ecosystem productivity, and precipitation’

Response: Changed.

regression between annual maximum NDVI, a proxy for ecosystem productivity, and vegetation greenness andprecipitation [6,11].

Line 71: change ‘objective’ to aim or purpose. This separates the overall aim of the work, from the specific objectives mentioned in line 74.

Response: Changed.

Theaimobjective of this study is to develop understanding of land degradation in the eastern…

Line 88: could add ‘as well as pollution from the power plants’

Response: Changed.

reducing surface and subsurface water resourcesas well as pollution from the power plants

Line 119: should this be ‘GIMMS NDVI3g’

Response: Changed.

change [32]. Monthly GIMMS NDVI was computed from the GMMIS-NDVI3gwith 16-day temporal

Line 125-126: delete ‘spatial distribution’ so text becomes ‘precipitation maps’

Response: Changed.

and temperatures from 1980 (http://cdcNaNa.gov.cn/home.do). The monthly precipitation spatial distributionmaps …

Line 131: change ‘from’ to ‘by’

Response: Yes, we followed your suggestion and changed.

Vegetation spatial distribution is shown in Figure 2. The dataset was originally producedbyfrom

Line 140: Change title to something like ‘Landsat processing in Google Earth Engine’ or ‘Validating breakpoint timing in Google Earth Engine’

Response: Changed.

2.2.4.Validatingbreakpoint timing inGoogle Earth Engine

Line 146: including the GEE code is good, but it would be nice if the code centred on the Region of Interest

Response: We changed the code with the function of ‘Map.setCenter()’.

Line 154: change ‘of’ to ‘for’

Response: Changed.

between peak growing season NDVI (NDVImax), a proxyforof NPP, and the optimal accumulated

Line 200: change ‘may indicate that there might have been significant structure change’ to ‘may indicate a significant structural change’

Response: Changed.

If a pixel has a significant breakpoint in VPR, it may indicateathat there might have beensignificant structurale change to the ecosystem during the study period [43].

Line 218: ‘Pixels where no significant model can be fitted are classified as indeterminate’

Response: Changed.

For pPixelswherethat no significant model can be fitted like aboveare classified as indeterminate.

Line 220: I think only formula 3 has been corrected for a small typo in Burrell et al., 2017, so the text should make this clear something like  ‘corrected a small typo in Burrell et al., 2017 that affected equation 3’.

Response: Changed.

Burrell et al. [8] gave detailed algorithm description in their paper with flow-chart; thereis a small typo in Burrell et al. [8] and we corrected this typo that affected equation 3.are some mistakes in formula in their paper, and here we corrected them (formulas 2, 3 and 4).

Line 226: I can’t see a definition for ‘NDVIts’ please add.

Response: Definition in line 224 added:

usingtime seriesNDVI(NDVIts)data

Line 259: change to ‘LTA, whereas TSS-RESTREND suggested degraded areas were distributed in the central and southwestern areas.’

Response: Changed.

LTA,whereas TSS-RESTREND suggested degradedareaswere distributed in the central and southeastern areas based onTSS-RESTREND

Page 12: sections 4.2 and 4.3 have the same heading.

Response: Changed heading of section 4.3 to land degradation.

4.3. Breakpoints detectionLand degradation
